

# Influence of tweets and diversification on serendipitous research paper recommender systems

Chifumi Nishioka[1,*], Jörn Hauke[2,*] and Ansgar Scherp[3]

[1] Kyoto University, Kyoto, Japan
[2] Christian-Albrechts-Universität Kiel, Kiel, Germany
[3] University of Essex, Colchester, UK
* These authors contributed equally to this work.

## ABSTRACT

In recent years, a large body of literature has accumulated around the topic of research paper recommender systems. However, since most studies have focused on the variable of accuracy, they have overlooked the serendipity of recommendations, which is an important determinant of user satisfaction. Serendipity is concerned with the relevance and unexpectedness of recommendations, and so serendipitous items are considered those which positively surprise users. The purpose of this article was to examine two key research questions: firstly, whether a user's Tweets can assist in generating more serendipitous recommendations; and secondly, whether the diversification of a list of recommended items further improves serendipity.

To investigate these issues, an online experiment was conducted in the domain of computer science with 22 subjects. As an evaluation metric, we use the serendipity score (SRDP), in which the unexpectedness of recommendations is inferred by using a primitive recommendation strategy. The results indicate that a user's Tweets do not improve serendipity, but they can reflect recent research interests and are typically heterogeneous. Contrastingly, diversification was found to lead to a greater number of serendipitous research paper recommendations.

## INTRODUCTION

To help researchers overcome the problem of information overload, various studies have developed recommender systems (*Beel et al., 2016*; *Bai et al., 2019*). Recommendations are generated based on considerations such as a user's own papers (*Sugiyama & Kan, 2010*; *Kaya, 2018*) or the papers a user has accessed or liked in the past (*Nascimento et al., 2011*; *Achakulvisut et al., 2016*). Most previous studies have focused only on improving the accuracy of recommendations, one example of which is normalised discounted cumulative gain (nDCG). However, several studies on recommender systems conducted in other domains (e.g. movies) have drawn attention to the fact that there are important aspects other than accuracy (*McNee, Riedl & Konstan, 2006*; *Herlocker et al., 2004*; *Kotkov, Wang & Veijalainen, 2016*; *Kotkov, Veijalainen & Wang, 2018*). One of these aspects is *serendipity*, which is concerned with the unexpectedness of recommendations and the

Corresponding author
Chifumi Nishioka,
nishioka.chifumi.2c@kyoto-u.ac.jp

degree to which recommendations positively surprise users (*Ge, Delgado-Battenfeld & Jannach, 2010*). A survey by *Uchiyama et al. (2011)* revealed that researchers think that it is important for them to be recommended serendipitous research papers.

In this article, we study a research paper recommender system focusing on serendipity. *Sugiyama & Kan (2015)* investigated serendipitous research paper recommendations, focusing on the influence of dissimilar users and the co-author network on recommendation performance. In contrast, this study investigates the following research questions:

- **(RQ1)** Do a user's Tweets generate serendipitous recommendations?
- **(RQ2)** Is it possible to improve a recommendation list's serendipity through diversification?

We run an online experiment to facilitate an empirical investigation of these two research questions using three factors. For RQ1, we employ the factor *User Profile Source*, where we compare the two sources of user profiles: firstly, a user's own papers; and secondly, a user's Tweets. The user's own papers are a feature of existing recommender systems, as evidenced by the work conducted by *Sugiyama & Kan (2015)* and Google Scholar (https://scholar.google.co.jp/). In this study, we assume that the user's Tweets produce recommendations that cannot be generated based on papers, since researchers Tweet about recent developments and interests that are yet not reflected in their papers (e. g. what they found interesting at a conference or in their social network) (*Letierce et al., 2010*). In addition, they are likely to have used their Twitter accounts to express private interests. We conjecture that taking private interests into consideration delivers serendipitous recommendations, since the recommender system will then suggest research papers that include both professional interests and private interests, and which are thus likely to be serendipitous. We also observed that recommendations based on a user's Tweets received a precision of 60%, which is fairly high in the domain of economics (*Nishioka & Scherp, 2016*).

Furthermore, we analyse the factor *Text Mining Method*, which applies different methods of candidate items (i.e. research papers) for computing profiles, as well as user profiles comprising different content (i.e. Tweets or previous papers).

As text mining methods, we compare TF-IDF (Salton & Buckley, 1988) with two of its recent extensions, namely CF-IDF (*Goossen et al., 2011*) and HCF-IDF (*Nishioka, Große-Bölting & Scherp, 2015*). Both have been associated with high levels of performance in recommendation tasks (*Goossen et al., 2011*; *Nishioka, Große-Bölting & Scherp, 2015*). We introduce this factor because text mining methods can have a substantial influence on generating recommendations. For RQ2, we introduce the factor *Ranking Method*, where we compare two ranking methods: firstly, classical cosine similarity; and secondly, the established diversification algorithm IA-Select (*Agrawal et al., 2009*). Cosine similarity has been widely used in recommender systems (*Lops, De Gemmis & Semeraro, 2011*), while IA-Select ranks candidate items with the objective of diversifying recommendations in a list. Since it broadens the coverage of topics in a list, we assume that IA-Select delivers more serendipitous recommendations compared to cosine similarity.

Along with the three factors *User Profile Source*, *Text Mining Method*, and *Ranking Method*, we conduct an online experiment in which 22 subjects receive research paper recommendations in the field of computer science. As an evaluation metric, we use the serendipity score (SRDP), which takes unexpectedness and usefulness of recommendations into account. It considers a recommendation as unexpected, if it is not recommended by a primitive recommendation strategy (i.e. baseline). The results reveal that a user's Tweets do not improve the serendipity of recommender systems. On the other hand, we confirm that the diversification of a recommendation list by IA-Select delivers more serendipitous recommendations to users.

The remainder of the paper is organised as follows: firstly, we describe related studies; in turn, we describe the recommender system and the experimental factors and evaluation setup; and finally, before concluding the article, we report on and discuss the experimental results.

## RELATED WORK

Over the last decade, many studies have developed research paper recommender systems (*Beel et al., 2016*; *Bai et al., 2019*). According to *Beel et al. (2016)*, more than half of these studies (55%) have applied a content-based approach. Collaborative filtering was applied by 18% and graph-based recommendations, utilising citation networks or co-authorship networks, were applied by 16%. Other researches have employed stereotyping, item-centric recommendations, and hybrid recommendations. In this article, we employ a content-based approach, as a number of works have done in the past with promising results (*Sugiyama & Kan, 2010*; *Nascimento et al., 2011*; *Achakulvisut et al., 2016*; *Kaya, 2018*).

### Clarifying the notion of serendipity

Most existing studies have evaluated research paper recommender systems by focusing on measures of accuracy, including precision, mean reciprocal rank (MRR), and normalised discounted cumulative gain (nDCG). However, studies that have addressed recommender systems in other domains (e.g. movies) argue that there are important considerations other than accuracy (*McNee, Riedl & Konstan, 2006*; *Herlocker et al., 2004*). One of these considerations is *serendipity*, which is a term that has been defined differently in the literature in the context of recommender systems. For instance, *Kotkov, Wang & Veijalainen (2016)* defined serendipity as "a property that indicates how good a recommender system is at suggesting serendipitous items that are relevant, novel and unexpected for a particular user." Similarly, *Herlocker et al. (2004)* defined serendipity as measure of the extent to which the recommended items are both attractive and surprising to the users. Other researchers have offered comparable definitions of serendipity (*Shani & Gunawardana, 2011*).

According to *Ge, Delgado-Battenfeld & Jannach (2010)*, it is important to recognise two important aspects of serendipity: firstly, a serendipitous item should be unknown to the user and, moreover, should not be expected; and secondly, the item should be interesting, relevant, and useful to the user. Taking these two aspects into account,

*Ge, Delgado-Battenfeld & Jannach (2010)* proposed a quantitative metric to evaluate the degree to which recommender systems are effective at generating serendipitous recommendations.

Most recently, *Kotkov et al. (2018)* conducted a literature review and operationalised common definitions of serendipity. Regarding unexpectedness, they organised four different definitions:

- Unexpectedness to be relevant (i.e. a user does not expect an item to be relevant).
- Unexpectedness to be found (i.e. a user would not have found this item on their own).
- Implicit unexpectedness (i.e. an item is significantly dissimilar to items a user usually consumes).
- Unexpectedness to be recommended (i.e. a user does not expect an item to be recommended).

In terms of novelty, they set two different definitions:

- Strict novelty (i.e. a user has never heard about an item or has consumed it and forgot about it).
- Motivationally novelty (i.e. a user has heard about an item, but has not consumed it).

This resulted in $4 \times 2 = 8$ definitions of serendipity. In addition, they investigated effects of different definitions of serendipity on preference broadening and user satisfaction. They found that all variations of unexpectedness and novelty broaden user preferences, but one variation of unexpectedness (unexpected to be relevant) hurts user satisfaction.

In this study, we evaluate the serendipity of recommendations using a metric proposed by *Ge, Delgado-Battenfeld & Jannach (2010)*. The metric considers a recommendation as unexpected, if it is not recommended by a primitive recommendation strategy (i.e. baseline). Thus, this study takes into account "unexpectedness to be found" and "unexpectedness to be recommended" in the different variations of unexpectedness proposed by *Kotkov et al. (2018)*. We choose this definition of serendipity as this is most relevant in our library context, where we recommend scientific papers to researchers (*Vagliano et al., 2018*).

## Use of social media for serendipitous recommendations

In previous studies addressing content-based research paper recommender systems (*Beel et al., 2016*; *Bai et al., 2019*), the authors calculated recommendations based on a user's own papers (*Sugiyama & Kan, 2010*) or papers a user has read in the past (*Nascimento et al., 2011*). In other domains, several studies have developed content-based recommender systems (*Chen et al., 2010*; *Orlandi, Breslin & Passant, 2012*; *Shen et al., 2013*) that utilise data from a user's social media accounts, including Twitter and Facebook. Another study proposed research paper recommendations based on a user's Tweets, which received a relatively high precision of 60% (*Nishioka & Scherp, 2016*). However, we hypothesise that because researchers Tweet about recent developments and interests that are not yet reflected in their papers (*Letierce et al., 2010*), a user's Tweets will deliver recommendations that are not generated based on papers.

In the context of research paper recommender systems, *Sugiyama & Kan (2015)* investigated serendipitous research paper recommendations focusing on the influence of dissimilar users and the co-author network on the recommendation performance. However, the researchers evaluated their approaches using accuracy-focused evaluation metrics such as nDCG and MRR. *Uchiyama et al. (2011)* considered serendipitous research papers as papers that are similar but in different fields from users' field. In contrast, this article investigates serendipitous research paper recommendations from the perspective of Tweets and diversification.

## Use of diversification for serendipitous recommendations

As discussed above, unexpectedness is a key concept for serendipity (*Ge, Delgado-Battenfeld & Jannach, 2010*). One approach that can be used to generate unexpected recommendations relates to diversification (*Ziegler et al., 2005*; *Agrawal et al., 2009*). This is because diversification leads to the creation of recommendation lists that include dissimilar items, meaning that users have an opportunity to encounter items they are unfamiliar with. IA-Select (*Agrawal et al., 2009*) has been used in the past as a solid baseline for diversifying lists of recommendations (*Vargas & Castells, 2011*; *Vargas, Castells & Vallet, 2011*; *Wu et al., 2018*). Additionally, Maximum Marginal Relevance (MMR) (*Carbonell & Goldstein, 1998*) is a well-known diversification method. *Kotkov, Veijalainen & Wang (2018)* proposed a serendipity-oriented greedy (SOG) algorithm, which diversifies a list of recommendations by considering unpopularity and dissimilarity. In this article, we employ IA-Select, because the experimental research conducted by *Vargas & Castells (2011)* shows that IA-Select performs better in general and the SOG algorithm requires a parameter setting.

# EXPERIMENTAL FACTORS

In this article, we build a content-based recommender system along with the three factors *User Profile Source*, *Text Mining Method*, and *Ranking Method*. It works as follows:

1. Candidate items of the recommender system (i.e. research papers) are processed by one of the text mining methods, and paper profiles are generated. A candidate item and a set of candidate items are referred as $d$ and $D$, respectively. $d$'s paper profile $P_d$ is represented by a set of features $F$ and their weights. Depending on text mining methods, a feature $f$ is either a textual term or a concept. Formally, paper profiles are described as: $P_d = \{(f, w(f, d)) \mid \forall f \in F \}$. The weighting function $w$ returns a weight of a feature $f$ for data source $I_u$. This weight identifies the importance of the feature $f$ for the user $u$.

2. A user profile is generated based on the user profile source (i.e. Tweets or own papers) using the same text mining method, which is applied to generate paper profiles. $I_u$ is a set of data items $i$ of a user $u$. In this article, $I_u$ is either a set of a user's Tweets or a set of a user's own papers. $u$'s user profile $P_u$ is represented in a way that it is comparable to $P_u$ as: $P_u = \{(f, w(f, I_u)) \mid \forall f \in F \}$.

3. One of the ranking methods determines the order of recommended papers.

**Table 1 Experimental factors and design choices.**

| Factor | Possible design choices | | |
|---|---|---|---|
| *User profile source* | Twitter | | Own papers |
| *Text mining method* | TF-IDF | CF-IDF | HCF-IDF |
| *Ranking method* | Cosine similarity | | IA-select |

The experimental design is illustrated in Table 1, where each cell is a possible design choice in each factor.

In this section, we first provide a detailed account of the factor *User Profile Source*. In turn, we show three of the different text mining methods that were applied in the experiment. Finally, we note the details of the factor *Ranking Method*, which examines whether diversification improves the serendipity of recommendations.

### User profile source

In this factor, we compare the following two data sources that are used to build a user profile.

- **Research papers:** The research papers written by a user are used as a baseline. This approach is motivated by previous studies that have investigated research paper recommender systems, including *Sugiyama & Kan (2010)* and Google Scholar.
- **Twitter:** In contrast to the user's papers, we assume that using Tweets leads to more serendipitous recommendations. It is common practice among researchers to Tweet about their professional interests (*Letierce et al., 2010*). Therefore, Tweets can be used to build a user profile in the context of a research paper recommender system. We hypothesise that a user's Tweets improve the serendipitous nature of recommendations because researchers are likely to Tweet about recent interests and information (e.g. from social networks) that are not yet reflected in their papers.

### Text mining method

For each of the two data sources (i.e. the user's own papers or their Tweets) and the candidate items, we apply a text mining method using one of three text mining methods. Specifically, we compare three methods, namely TF-IDF (*Salton & Buckley, 1988*), CF-IDF (*Goossen et al., 2011*), and HCF-IDF (*Nishioka, Große-Bölting & Scherp, 2015*), to build paper profiles and a user profile. This factor was introduced because the effectiveness of each text mining method is informed by the type of content that will be analysed (e.g. Tweets or research papers). For each method, a weighting function $w$ is defined. This weighting function assigns a specific weight to each feature $f$, which is a term in TF-IDF and a semantic concept in CF-IDF and HCF-IDF.

- **TF-IDF:** Since TF-IDF is frequently used in recommender systems as a baseline (*Goossen et al., 2011*), we also use it in this study. Terms are lemmatised and stop words are removed (http://www.nltk.org/book/ch02.html). In addition, terms with fewer than

three characters are filtered out due to ambiguity. After pre-processing texts, TF-IDF is computed as:

$$w_{tf\text{-}idf}(w, t) = tf(w, t) \cdot \log \frac{|D|}{|\{w \in d : d \in D|\}} \tag{1}$$

*tf* returns the frequency of a term *w* in a text *t*. A text *t* is either a user profile source $I_u$ or candidate item *d*. The term frequency acts under the assumption that more frequent terms are more important (*Salton & Buckley, 1988*). The second term of the equation presents the inverse document frequency, which measures the relative importance of a term *w* in a corpus *D* (i.e. a set of candidate items).

- **CF-IDF:** Concept frequency inverse document frequency (CF-IDF) (*Goossen et al., 2011*) is an extension of TF-IDF, which replaces terms with semantic concepts from a knowledge base. The use of a knowledge base decreases noise in profiles (*Abel, Herder & Krause, 2011*; *Middleton, Shadbolt & De Roure, 2004*). In addition, since a knowledge base can store multiple labels for a concept, the method directly supports synonyms. For example, the concept "recommender systems" of the ACM Computing Classification Systems (ACM CCS) has multiple labels, including "recommendation systems", "recommendation engine", and "recommendation platforms".

  The weighting function *w* for CF-IDF is defined as:

$$w_{cf\text{-}idf}(a, t) = cf(a, t) \cdot \log \frac{|D|}{|\{a \in d : d \in D|\}} \tag{2}$$

  *cf* returns the frequency of a semantic concept *a* in a text *t*. The second term presents the IDF, which measures the relative importance of a semantic concept *a* in a corpus *D*.

- **HCF-IDF:** Finally, we apply hierarchical concept frequency inverse document frequency (HCF-IDF) (*Nishioka, Große-Bölting & Scherp, 2015*), which is an extension of CF-IDF. HCF-IDF applies a propagation function (*Kapanipathi et al., 2014*) over a hierarchical structure of a knowledge base to assign a weight to concepts at higher levels. In this way, it identifies concepts that are not mentioned in a text but which are highly relevant. HCF-IDF calculates the weight of a semantic concept *a* in a text *t* as follows:

$$w_{hcf\text{-}idf}(a, t) = BL(a, t) \cdot \log \frac{|D|}{|\{d \in D : a \in d\}|} \tag{3}$$

  $BL(a, t)$ is the BellLog propagation function (*Kapanipathi et al., 2014*), which is defined as:

$$BL(a, t) = cf(a, t) + FL(a) \cdot \sum_{a_j \in pc(a)} BL(a_j, t) \tag{4}$$

  where $cf(a, t)$ is a frequency of a concept *a* in a text *t*, and $FL(a) = \dfrac{1}{\log_{10}(nodes(h(a) + 1))}$. The propagation function underlies the assumption that, in human memory, information is represented through associations or semantic

networks (*Collins & Loftus, 1975*). The function $h(a)$ returns the level, where a concept $a$ is located in the knowledge base. Additionally, *nodes* provides the number of concepts at a given level in a knowledge base, and $pc(a)$ returns all parent concepts of a concept $a$. In this study, we employ HCF-IDF since it has been shown to work effectively for short pieces of text, including Tweets (*Nishioka & Scherp, 2016*), in the domain of economics.

### Ranking method

Finally, we rank all the candidate items to determine which items should be recommended to a user. In this factor, we compare two ranking methods: cosine similarity and diversification with IA-Select (*Agrawal et al., 2009*).

- **Cosine similarity:** As a baseline, we employ a cosine similarity, which has been widely used in content-based recommender systems. The top-$k$ items with largest cosine similarities are recommended.

- **IA-Select:** Following this, we employ IA-Select (*Agrawal et al., 2009*) to deliver serendipitous recommendations. IA-Select was originally introduced for information retrieval, but it is also used in recommender systems to improve serendipity (*Vargas, Castells & Vallet, 2012*). This use case stems from the algorithm's ability to diversify recommendations in a list, which relies on the avoidance of recommending similar items (e.g. research papers) together. The basic idea of IA-Select is that, for those features of a user profile that have been covered by papers already selected for recommendation, the weights are lowered in an iterative manner. At the outset, the algorithm computes cosine similarities between a user and each candidate item. In turn, IA-Select adds the item with the largest cosine similarity to the recommendation list. After selecting the item, IA-Select decreases the weights of features covered by the selected item in the user profile. These steps are repeated until $k$ recommendations are determined.

For example, recommendations for the user profile $P_u = ((f_1, 0.1), (f_2, 0.9))$ will contain mostly those documents that include feature $f_2$. However, with IA-Select, the $f_2$ score is decremented iteratively in the event that documents contain the $f_2$ feature. Thus, the probability increases that documents covering the $f_1$ feature are included in the list of recommended items.

Overall, the three factors with the design choices described above result in $2 \times 3 \times 2 = 12$ available strategies. The evaluation procedure used to compare the strategies is provided below.

## EVALUATION

To address the two research questions with the three experimental factors described in the previous section, we conduct an online experiment with 22 subjects. The experiment is based in the field of computer science, in which an open access culture to research papers exists, and Twitter is chosen as the focal point because it is an established means by which researchers disseminate their works. The experimental design adopted in this study is consistent with previous studies (*Nishioka & Scherp, 2016*; *Chen et al., 2010*).

**Recommendation (1/12)**

Please evaluate the following randomized list of the top five publications "interesting" or "not interesting".
Click on a title to see its abstract in a new window.

**Please Note:** The list might contain publications which you have already seen, since the system makes recommendations under different, independent strategies.

- Robin J. Wilson, "Stamps, computing on", Encyclopedia of Computer Science, 2003 — ○ interesting ○ not interesting
- Sven Uebelacker, Susanne Quiel, "The Social Engineering Personality Framework", STAST '14 Proceedings of the 2014 Workshop on Socio-Technical Aspects in Security and Trust, 2014 — ○ interesting ○ not interesting
- Katharina Krombholz, Heidelinde Hobel, Markus Huber, Edgar Weippl, "Social engineering attacks on the knowledge worker", Proceedings of the 6th International Conference on Security of Information and Networks, 2013 — ○ interesting ○ not interesting
- Michael Workman, "Gaining Access with Social Engineering: An Empirical Study of the Threat", Information Systems Security, 2007 — ○ interesting ○ not interesting
- Anker Helms Jørgensen, Brad A. Myers, "User interface history", CHI '08 Extended Abstracts on Human Factors in Computing Systems, 2008 — ○ interesting ○ not interesting

**Figure 1** **Screenshot of the evaluation page.** Each subject rated an item as either "interesting" or "not interesting" based on their research interests.

In this section, the experimental design is described, after which an account of the utilised datasets (i.e. a corpus of research papers and a knowledge graph of text mining methods) is given. Following this, descriptive statistics are presented for the research subjects, and finally, the serendipity score is stated. The purpose of the serendipity score is to evaluate the degree to which each recommender strategy is effective in generating serendipitous recommendations.

## Procedure

We implemented a web application that enabled the subjects ($n = 22$) to evaluate the 12 recommendation strategies described above. First, subjects started on the welcome page, which asked for their consent to collect their data. Thereafter, the subjects were asked to input their Twitter handle and their name, as recorded in DBLP Persons (https://dblp.uni-trier.de/pers/). Based on the user's name, we retrieved a list of their research papers and obtained the content of the papers by mapping them to the ACM-Citation-Network V8 dataset (see below). The top 5 recommendations were computed for each strategy, as shown in Fig. 1. Thus, each subject evaluated $5 \times 12 = 60$ items as "interesting" or "not interesting" based on the perceived relevance to their research interests.

A binary evaluation was chosen to minimise the effort of the rating process, consistent with several previous studies (*Nishioka & Scherp, 2016*; *Chen et al., 2010*). As shown in Fig. 1, the recommended items were displayed with bibliographic information such as the authors, title, year, and venue. Finally, the subjects were provided with the opportunity to access and read the research paper directly by clicking on a link. In order to avoid bias, the sequence in which the 12 strategies appeared was randomised for each subject. This corresponds to earlier experimental setups such as a research paper recommender system in the domain of economics (*Nishioka & Scherp, 2016*) and other studies (*Chen et al., 2010*). At the same time, the list of the top 5 items for each strategy was also randomised to avoid the well-documented phenomenon of ranking bias (*Bostandjiev, O'Donovan & Höllerer, 2012*;

*Chen et al., 2010*). The subjects were informed about the randomised order of the strategies and items on the evaluation page.

The actual ranks of the recommended items, as well as their position on the evaluation page, were stored in a database for later analyses. After evaluating all strategies, the subjects were asked to complete a questionnaire focusing on demographic information (e.g. age, profession, highest academic degree, and current employment status). Finally, an opportunity was provided for the subjects to provide qualitative feedback.

## Datasets

The candidate items for the experiment were computer science articles drawn from a large dataset of research papers. To analyse and extract semantic concepts from the research papers and Tweets, an external computer science knowledge base was used. This section describes the research papers and knowledge graphs used for the experiment.

### Research papers

Since the experiment recommended research papers from the field of computer science, a corpus of research papers and a knowledge base from the same field were used. The ACM citation network V8 dataset (https://lfs.aminer.org/lab-datasets/citation/citation-acm-v8.txt.tgz), provided by ArnetMiner (*Tang et al., 2008*), was used as the corpus of research papers. From the dataset, 1,669,237 of the available 2,381,688 research papers were included that had a title, author, year of publication, venue, and abstract. Titles and abstracts were used to generate paper profiles.

### Knowledge graph

The ACM Computing Classification System (CCS) was used as the knowledge graph for CF-IDF and HCF-IDF (https://www.acm.org/publications/class-2012). The knowledge graph, which is freely available, is characterised by its focus on computer science, as well as its hierarchical structure. It consists of 2,299 concepts and 9,054 labels, which are organised on six levels. On average, a concept is represented by 3.94 labels (SD: 3.49).

For the text mining methods (i.e. CF-IDF and HCF-IDF), we extracted concepts from each user's Tweets and research papers by matching the text with the labels of the concepts in the knowledge graph. As such, we applied what is known in the literature as the gazetteer-based approach. Before processing, we lemmatised both the Tweets and research papers using Stanford Core NLP (https://stanfordnlp.github.io/CoreNLP/), and stop words were removed. Regarding Tweets, which often contain hashtags to indicate topics and user mentions, only the # and @ symbols were removed from the Tweets. This decision stemmed from an observation made by *Feng & Wang (2014)*, namely that the combination of Tweets' texts with hashtags and user mentions results in the optimal recommendation performance.

## Subjects

Overall, 22 subjects were recruited through Twitter and mailing lists. 20 were male and two were female, and the average age was 36.45 years old (SD: 5.55). Several of the subjects held master's degrees ($n = 2$), while the others held a PhD ($n = 13$) or were lecturers or

**Table 2 The number of subjects in each country.**

| Country | The number of subjects |
| --- | --- |
| Germany | 8 |
| US | 4 |
| China | 2 |
| UK | 2 |
| Austria | 1 |
| Brazil | 1 |
| France | 1 |
| Ireland | 1 |
| Norway | 1 |
| Sweden | 1 |

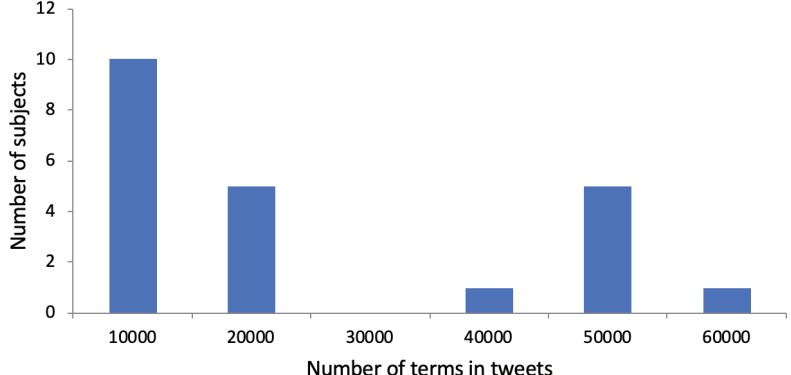

**Figure 2 Distribution of subjects with regarding to the number of terms in their Tweets.** The $x$-axis shows the number of terms in their Tweets. The $y$-axis shows the number of subjects. For instance, there are five subjects whose total number of terms in Tweets is between 10,001 and 20,000.

professors ($n = 7$). In terms of the subjects' employment status, 19 were working in academia and three in industry. Table 2 shows countries where subjects work. On average, the subjects published 1,256.97 Tweets (SD: 1,155.8), with the minimum value being 26 and the maximum value being 3,158.

An average of 4,968.03 terms (SD: 4,641.76) was extracted from the Tweets, along with an average of 297.91 concepts (SD: 271.88). Thus, on average, 3.95 (SD: 0.54) terms and 0.24 concepts (SD: 0.10) were included per Tweet. We show a histogram regarding the number of terms in Tweets per subject in Fig. 2. We observe that subjects are divided into those with a small total number of terms in their Tweets and those with a large total number of terms in their Tweets. Regarding the use of research papers for user profiling, the subjects had published an average of 11.41 papers (SD: 13.53). On average, 687.68 terms (SD: 842,52) and 80.23 concepts (SD: 107.73) were identified in their research papers. This led to 60.27 terms (SD: 18.95) and 5.77 concepts (SD: 3.59) per paper. Figure 3 shows a histogram regarding the number of terms in research papers per subject. We see

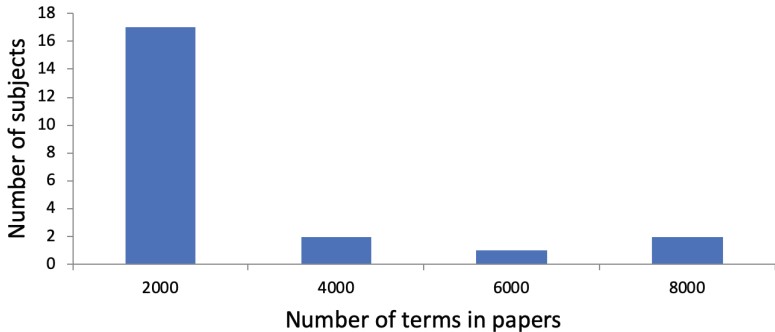

**Figure 3 Distribution of subjects with regarding to the number of terms in their research papers.** The *x*-axis shows the number of terms in their research papers. The *y*-axis shows the number of subjects. For instance, there are two subjects whose total number of terms in research papers is between 2,001 and 4,000.

that there are a few subjects with a large total number of terms. Most subjects have a small total number of terms in their research papers because they published only a few research papers so far.

Subjects needed 39 s (SD: 43 s) on average to evaluate all five recommended items per strategy. Thus, the average length of time needed to complete the experiment was 468 s. It is worth noting that this time does not include reading the instructions on the welcome page, inputting the Twitter handle and DBLP record, and completing the questionnaire.

## Evaluation metric

To evaluate the serendipity of recommendations, we used the serendipity score (SRDP) (*Ge, Delgado-Battenfeld & Jannach, 2010*). This evaluation metric takes into account both the unexpectedness and usefulness of recommended items, which is defined as:

$$\text{SRDP} = \sum_{d \in UE} \frac{rate(d)}{|UE|} \tag{5}$$

*UE* denotes a set of unexpected items that are recommended to a user. An item is regarded as unexpected if it is not included in a recommendation list computed by the primitive strategy. We used the strategy Own Papers × TF-IDF × Cosine Similarity as a primitive strategy since it is a combination of baselines. The function *rate(d)* returns an evaluation rate of an item *d* given by a subject. As such, if a subject evaluated an item as "interesting", the function would return 1, otherwise 0.

We did not directly ask subjects to evaluate the unexpectedness of recommendations, because this is not the scenario in which the recommender system is used. Rather, we were aiming to detect indirectly from the subjects' responses, if the serendipity feature had an influence on the dependent variables. Furthermore, we wanted to keep the online evaluation as simple as possible. Asking for "how surprising" a recommendation is, increases the complexity of the experiment. Subjects needed to know what a non-surprising recommendation is (in comparison). In addition, the cognitive efforts

**Table 3 SRDP and the number of unexpected items included in the 12 strategies.** The values are ordered by SRDP. M and SD denote mean and standard deviation, respectively.

| | Strategy | | | SRDP | \|UE\| |
|---|---|---|---|---|---|
| | Text mining method | Profiling source | Ranking method | M (SD) | M (SD) |
| 1. | TF-IDF | Own papers | IA-Select | 0.45 (0.38) | 2.95 (1.05) |
| 2. | CF-IDF | Twitter | CosSim | 0.39 (0.31) | 4.91 (0.29) |
| 3. | TF-IDF | Twitter | IA-Select | 0.36 (0.29) | 4.91 (0.43) |
| 4. | CF-IDF | Twitter | IA-Select | 0.31 (0.22) | 4.95 (0.21) |
| 5. | CF-IDF | Own papers | CosSim | 0.26 (0.28) | 4.91 (0.29) |
| 6. | CF-IDF | Own papers | IA-Select | 0.25 (0.28) | 4.91 (0.29) |
| 7. | HCF-IDF | Own papers | IA-Select | 0.24 (0.22) | 4.95 (0.21) |
| 8. | HCF-IDF | Twitter | CosSim | 0.22 (0.28) | 5.00 (0.00) |
| 9. | TF-IDF | Twitter | CosSim | 0.20 (0.24) | 4.95 (0.21) |
| 10. | HCF-IDF | Twitter | IA-Select | 0.18 (0.21) | 5.00 (0.00) |
| 11. | HCF-IDF | Own papers | CosSim | 0.16 (0.18) | 5.00 (0.00) |
| 12. | TF-IDF | Own papers | CosSim | 0.00 (0.00) | 0.00 (0.00) |

required to conduct a direct evaluation of unexpectedness is much higher and it is in general difficult for subjects to share the concept of the unexpectedness.

## RESULTS

The purpose of this section is to present the results of the experiment. At the outset, the quantitative analysis is examined, which shows the optimal strategy in terms of SRDP. In turn, the impact of each of the three experimental factors is analysed.

### Comparison of the 12 strategies

The results of the 12 strategies in terms of their SRDP values are presented in Table 3. As previously noted, this study drew on Own Papers × TF-IDF × Cosine Similarity as a primitive strategy. Thus, for this particular strategy, the mean and standard deviation are 0.00.

The purpose of an analysis of variance (ANOVA) is to detect significant differences between variables. Therefore, in this study, ANOVA was used to identify whether any of the strategies were significantly different. The significance level was set to $\alpha = 0.05$. Mauchly's test revealed a violation of sphericity ($\chi^2(54) = 80.912$, $p = 0.01$), which could lead to positively biased $F$-statistics and, consequently, an increase in the risk of false positives. Therefore, a Greenhouse-Geisser correction with $\varepsilon = 0.58$ was applied.

The results of the ANOVA test revealed that significant differences existed between the strategies ($F(5.85, 122.75) = 3.51$, $p = 0.00$). Therefore, Shaffer's modified sequentially rejective Bonferroni procedure was undertaken to compute the pairwise differences between the strategies (*Shaffer, 1986*). We observed significant differences between the primitive strategy and one of the other strategies.

**Table 4 Three-way repeated measures ANOVA for SRDP with Greenhouse-Geisser correction and F-ratio, effect size $\eta^2$, and p-value.** The p-values are marked in bold font if $p < .05$, which indicates a significant difference in a factor.

| Factor | F | $\eta^2$ | p |
|---|---|---|---|
| *User profile source* | 2.21 | 0.11 | 0.15 |
| *Text mining method* | 3.02 | 0.14 | 0.06 |
| *Ranking method* | 14.06 | 0.67 | **0.00** |
| *User profile source × Text mining method* | 0.98 | 0.05 | 0.38 |
| *User profile source × Ranking method* | 18.20 | 0.87 | **0.00** |
| *Text mining method × Ranking method* | 17.80 | 0.85 | **0.00** |
| *User profile source × Text mining M. × Ranking M.* | 2.39 | 0.11 | 0.11 |

## Impact of experimental factors

In order to analyse the impact of each experimental factor, a three-way repeated measures ANOVA was conducted. The Mendoza test identified violations of sphericity for the following factors: firstly, *User Profile Source × Text Mining Method × Ranking Method* ($\chi^2(65) = 101.83$, $p = 0.0039$); and secondly, *Text Mining Method × Ranking Method* ($\chi^2(2) = 12.01$, $p = 0.0025$) (*Mendoza, 1980*). Thus, a three-way repeated measures ANOVA was applied with a Greenhouse-Geiser correction of $\varepsilon = 0.54$ for the factors *User Profile Source × Text Mining Method × Ranking Method* and $\varepsilon = 0.69$ for the factor *Text Mining Method × Ranking Method*. Table 4 shows the results with the *F*-Ratio, effect size $\eta^2$, and p-value.

Regarding the single factors, *Ranking Method* had the largest impact on SRDP, as the effect size $\eta^2$ indicates. For all the factors with significant differences, we applied a post-hoc analysis using Shaffer's MSRB procedure. The results of the post-hoc analysis revealed that the strategies using IA-Select resulted in higher SRDP values when compared to those using cosine similarity. In addition, we observed a significant difference in the factors *User Profile Source × Ranking Method* and *Text Mining Method × Ranking Method*. For both factors, post-hoc analyses revealed significant differences when a baseline was used in either of the two factors. When a baseline was used in one factor, |UE| became small unless a method other than a baseline was used in the other factor.

## DISCUSSION

This section discusses the study's results in relation to the two research questions. In turn, we review the results for the *Text Mining Method* factor, which was found to have the largest influence on recommendation performance among the three factors.

**RQ1**: Do a user's Tweets generate serendipitous recommendations?

Regarding RQ1, the results of the experiment indicate that a user's Tweets do not improve the serendipity of recommendations. As shown in the rightmost column of Table 3, Tweets deliver unexpected recommendations to users, but only a small fraction of these are interesting to the users. This result is different from previous works. For instance, *Chen et al. (2010)* observed the precision of a webpage recommender system based on

user's Tweets was around 0.7. In addition, *Lu, Lam & Zhang (2012)* showed that a concept-based tweet recommender system based on user's Tweets achieves a precision of 0.5. One way to account for this result is by drawing attention to the high probability that the users employed their Twitter accounts for purposes other than professional, research-related ones. In particular, the users are likely to have used their Twitter accounts to express private interests. We presume that taking private interests into consideration delivers serendipitous recommendations. This is because the recommender system will then suggest research papers that include both professional interests and private interests, and which are thus likely to be serendipitous. In the future, it may be helpful to introduce explanation interfaces for recommender systems (*Herlocker, Konstan & Riedl, 2000*; *Tintarev & Masthoff, 2007*). The purpose of these explanation interfaces is to show why a specific item is being recommended to users, thereby enabling users to find a connexion between a recommended paper and their interests.

**RQ2**: Is it possible to improve a recommendation list's serendipity through diversification?

In terms of RQ2, the results indicate that the diversification of a recommendation list using the IA-Select algorithm delivers serendipitous recommendations. This confirms results published elsewhere in the literature, which have found that IA-Select improves serendipity (*Vargas, Castells & Vallet, 2011*; *Vargas & Castells, 2011*). For instance, in the domain of movies and music, *Vargas & Castells (2011)* employed IA-Select for recommender systems and confirmed that it provides unexpected recommendations. Additionally, the iterative decrease of covered interests was associated with greater variety in recommender systems for scientific publications. Furthermore, the experiment demonstrated that diversified recommendations are likely to be associated with greater utility for users.

## Text mining methods

Among the three factors, the *Text Mining Method* factor was associated with the most substantial impact on recommender system performance. In contrast to observations made in previous literature (*Goossen et al., 2011*; *Nishioka & Scherp, 2016*), CF-IDF and HCF-IDF did not yield effective results. It is worth emphasising that this result could have been influenced by the quality of the knowledge graph used in this study (i.e. ACM CCS), particularly in view of the fact that the performance of many text mining methods is directly informed by the quality of the knowledge graph (*Nishioka, Große-Bölting & Scherp, 2015*).

Another way to account for the poor outcomes relates to the variable of the knowledge graphs' age. In particular, ACM CCS has not been updated since 2012, despite the fact that computer science is a rapidly changing field of inquiry. Furthermore, relatively few concepts and labels were included in the knowledge base, which contrasts with the large number included in the knowledge graphs used in previous studies. For example, the STW Thesaurus for Economics used 6,335 concepts and 37,773 labels, respectively (*Nishioka & Scherp, 2016*). Hence, the number of concepts and labels could have

influenced the quality of the knowledge graph and, in turn, the recommender system's performance.

In addition, while a previous study that used HCF-IDF (*Nishioka & Scherp, 2016*) only drew on the titles of research papers, our study used both titles and abstracts to construct paper profiles and user profiles when a user's own papers were selected as the user profile source. Furthermore, since our study used sufficient information when mining research papers, we did not observe any differences among TF-IDF, CF-IDF, and HCF-IDF, which can include related concepts. Finally, as with any empirical experiment, data triangulation is needed before generalising any of the conclusions drawn in this paper. Therefore, further studies of recommender systems in other domains and similar settings should be conducted.

In this article, we used only textual information in Tweets. We did not use contents from URLs mentioned in Tweets, images, and videos. We observed that Tweets by subjects contain on average 0.52 URLs (SD: 0.59). In the future, we would like to take these contents into account, as *Abel et al. (2011)* did.

### Threats to validity

In this article, we only considered the domain of computer science. In other domains, the results and findings might be different. In the future, we would like to conduct studies in other domains such as biomedical science using MEDLINE and social science, economics. In addition, the results shown in this article may potentially be influenced by the number of subjects we recruited. Finding significances with few subjects is harder than with many subjects. However, we observed several significances and measured the effect sizes. We assume that adding more subjects would bring almost no additional insights.

As noted in the related work, this study evaluates serendipity of recommendations focusing on "unexpectedness to be found" and "unexpectedness to be recommended". This is motivated by our library setting, where we assume researchers are working on research papers and like to receive recommendations for literature that they have not found yet (*Vagliano et al., 2018*). Referring to the other variations of serendipity as proposed by *Kotkov et al. (2018)*, like "unexpectedness to be relevant" and "implicit unexpectedness", we leave them for future studies.

## CONCLUSION

The purpose of this study's online experiment was to determine whether Tweets and the IA-Select algorithm have the capability to deliver serendipitous research paper recommendations. The results revealed that Tweets do not improve the serendipity of recommendations, but IA-Select does. We anticipate that this insight will contribute to the development of future recommender systems, principally because service providers and platform administrators can use the data presented here to make more informed design choices for the systems and services developed. The data from this experiment are publicly available for further study and reuse (https://doi.org/10.5281/zenodo.3367795).

### Funding

This work was supported by the EU H2020 project MOVING (No. 693092), the JSPS Grant-in-Aidfor Scientific Research (S) (No. 16H06304), and the JSPS Grant-in-Aid for Young Scientists (No. 18K13235). The funders had no role in study design, data collection and analysis, decision to publish, or preparation of the manuscript.

### Grant Disclosures

The following grant information was disclosed by the authors:
EU H2020 project MOVING: 693092.
JSPS Grant-in-Aid for Scientific Research (S): 16H06304.
JSPS Grant-in-Aid for Young Scientists: 18K13235.

### Competing Interests

The authors declare that they have no competing interests.

### Author Contributions

- Chifumi Nishioka conceived and designed the experiments, analysed the data, performed the computation work, prepared figures and/or tables, authored or reviewed drafts of the paper, and approved the final draft.
- Jörn Hauke conceived and designed the experiments, performed the experiments, analysed the data, performed the computation work, prepared figures and/or tables, and approved the final draft.
- Ansgar Scherp conceived and designed the experiments, analysed the data, authored or reviewed drafts of the paper, and approved the final draft.

### Data Availability

The data is available at Zenodo: Nishioka, Chifumi, Scherp, Ansgar, & Hauke, Jörn. (2019). Experimental result to investigate the influence of user's tweets and diversification on serendipitous research paper recommendations [Data set]. Zenodo. DOI 10.5281/zenodo.3367796.

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
