# Peer review of "Influence of tweets and diversification on serendipitous research paper recommender systems"

_PeerJ Computer Science, doi:10.7717/peerj-cs.273_

## Round 0.1 · original submission · Major Revisions

Myself and two reviewers have evaluated your paper. Please address the concerns of each reviewer. In addition, I would like to know why only 2 women were included in the study. Also, where are these users located - a country break down is needed.

Reviewer 1 ·

Basic reporting

The article investigates the effect of tweets and diversification on serendipity of recommender systems in the domain of research articles. The authors motivated their study with literature review, conducted a user study with 22 subjects, which is valued in the field of recommender systems and analyzed the results with rigorous statistical methods. The article is well structured and easy to ready, but it has a number of drawbacks.
1. It would be better to bring the argument regarding researchers’ private interests page 9 (lines 322 - 327) in the introduction (introduction only discusses tweets on research interests).

Experimental design

1. The main drawback of the study is the way the experiment was designed. The authors asked users to indicate interesting articles and artificially calculated serendipity (eq. 5) with the recommendation list generated by the primitive recommender system (p. 7, line 275). Meanwhile, the authors could directly ask users if they find a particular article serendipitous.
Please indicate explicitly in introduction and abstract that serendipity has been calculated artificially (based on primitive recommender system) and not by asking users.

Validity of the findings

1. The way serendipity was calculated in equation 5 (p. 7) does not correspond to the definition of serendipity given in the abstract, as this equation does not take into account novelty of the article (users were not asked about novelty of articles either). Please either change the definition of serendipity used throughout the article or change the way it is calculated.
2. RQ2 is missing an important reference, which discusses the same research question in the movie domain:
Kotkov, D., Veijalainen, J., & Wang, S. (2018). How does serendipity affect diversity in recommender systems? A serendipity-oriented greedy algorithm. Computing, 1-19.
The discussion section should include comparison of the obtained results with the results obtained by other studies.

Additional comments

Overall, the study is well reported and contains contributions important for the field. After the mentioned drawbacks have been corrected, the article can become a good candidate for publication.

·

Basic reporting

In this paper, the authors explore the serendipity of the paper recommendation system. The serendipity can be calculated by a summation of the rating divide by the number of unexpected items of the recommendations. The authors apply a variety of content-based recommendation systems using publication content and twitter to recommend publications. Content used for recommendation can be selected from the author's paper or twitter. The recommendation is calculated using multiple calculation techniques such as TF-IDF, CF-IDF, and HCF-IDF. And the distance metric is selected from cosine distance (rank by similarity) and IA-select (to diversify the search result).

They aim to (1) find the impact of twitter on a content-based recommendation while measure the serendipity score outcome and (2) see if algorithms like IA-select which can diversify the search result can improve serendipitous recommendation.

Overall, the paper explains the task and result quite well. However, it lacks quite a bit of a literature review in the introduction section. For example, there are a papers that present content-based recommendations such as Science Concierge (https://journals.plos.org/plosone/article?id=10.1371/journal.pone.0158423) and User Profile Based Paper Recommendation (https://ijisae.org/IJISAE/article/view/752). Lastly, adding more details in the experimental section can improve the paper. For example, I would like the author to add one a section on how do they present the recommendation result sequentially to the subject i.e. which strategy from 1 to 12 gets presented first to last.

My main criticism of the paper is the usage of twitter profiles in the experiment. It can be the case that the users tweet about the publication with relatively short text with and extra of publication's URL. I assume that if all tweet text and content from the tweet URL are included in the recommendation engine, it might give a better result for twitter. This is a little hard to re-experiment but it can be noted in the discussion section.

Experimental design

- The subjects are well recruited. However, I think that the subject size is a little bit too small (n = 22) in the experiment given that the task can be finished relatively quickly in less than 10 minutes.
- As mentioned above, I would like the author to add one section on how do they present the recommendation result sequentially to the subject i.e. which strategy from 1 to 12 gets presented first to last.
- The dataset could include concepts extracted from twitter and the author can also visualize using the histogram for the length of the text from Twitter. It is a bit hard to know the distribution of tweets length in all the subjects seeing just average length and its standard deviation.

Validity of the findings

The finding is very interesting and novel in a sense of outcome measurement of serendipity. I agree that the usage of twitter alone might not help improve the recommendation engine much.

The current number of subjects is quite small. If possible, by increasing the number of subjects, it will make the paper much stronger.

The authors also only explore the publications in the computer science field. I think it would be good if they can clarify in the discussion section that it has to explore more on different fields such as biomedical science (Pubmed/MEDLINE), social science, or economics.

---

## Round 0.2 · Minor Revisions

Thank you very much for your revision. Please address the final comments from reviewer #1.

Reviewer 1 ·

Basic reporting

For instance, Kotkov et al.  (2016) defined serendipity as “a property that indicates how good a recommender system is at suggesting serendipitous items that are relevant and unexpected for a particular user.” - the direct quotation does not correspond to the original text of the cited article.
The authors should provide arguments for choosing this particular definition of serendipity.
There is a study that provides the effects of different definitions of serendipity on users:
Kotkov D. et al. Investigating serendipity in recommender systems based on real user feedback //Proceedings of the 33rd Annual ACM Symposium on Applied Computing. – ACM, 2018. – С. 1341-1350.

Experimental design

Experimental design is well explained

Validity of the findings

The findings are valid

---

## Round 0.3 · accepted · Accept

Thank you for your response to the referee comments from the previous round of reviews.